# Communication and Interaction Practices in Czech Classrooms with a Teaching Assistant

**Martina Maněnová**, **Janet Wolf**, **Martin Skutil** * and **Jitka Vítová**

Institute of Primary, Pre-Primary and Special Education, Faculty of Education, University of Hradec Králové, Rokitanskeho 62, 500 03 Hradec Králové, Czech Republic; martina.manenova@uhk.cz (M.M.); janet.wolf@uhk.cz (J.W.); jitka.vitova@uhk.cz (J.V.)
* Correspondence: martin.skutil@uhk.cz; Tel.: +420-777-593-912

**Abstract:** The presented study focuses on pedagogical communication and interaction occurring in teaching with the presence of a teaching assistant. The aim was to enhance understanding of pedagogical communication and interaction in these classrooms. To achieve this goal, an analysis of teaching sessions was undertaken from the perspective of activities (interactions and communication) assumed by the teaching assistant during instruction. The research methodology relies on standardized observation, specifically employing the Flanders Interaction Analysis System, which investigates communication and interaction within the classroom environment. For observing teaching sessions with teaching assistants, 16 categories were developed and refined through pilot testing. Validation was conducted using video analysis. In total, 15 teaching sessions were recorded and subsequently analyzed. Specialized software Codenet was utilized for video data analysis, with a set time interval of 3 s. The data analysis revealed several crucial insights into the communication and interaction of teaching assistants and teachers in an educational context. Some categories, such as the preparation of teaching aids and those related to passivity, predominate, while others, including active involvement in teaching or introducing new educational content, are notably absent. Understanding these patterns may lead to optimizing the role of the teaching assistant in supporting individual students and classroom dynamics. The article discusses inclusive practices of teaching assistants, without distinguishing between students with special educational needs and typically developing peers. Inclusive education promotes social and environmental sustainability by fostering a sense of belonging and equality among students.

**Keywords:** teaching assistant; teacher; communication; interaction; FIAS

## 1. Introduction

Communication serves as the fundamental driving force in any relational or situational context. In educational settings, teachers utilize communication to achieve three primary objectives: to extract relevant knowledge from students, to respond effectively to students' expressions, and to articulate shared classroom experiences [1]. Communication represents an ongoing process of transmitting and receiving messages, facilitating the exchange of knowledge, attitudes, and skills among individuals. Effective teaching is contingent upon successful communication, with explicit communication occurring when teachers and students interact [2].

The teaching profession demands impeccable communication skills, as a teacher's capacity to foster student development hinges significantly on effective communication [3]. Consequently, teachers must excel both in interpersonal communication and in executing technical tasks to attain professional success.

The communication styles adopted by teachers play a pivotal role in assisting students in harmonizing their self-regulation with intrinsic motivation during autonomous classroom activities [4]. The establishment of effective classroom interaction heavily relies on communication, encompassing verbal and non-verbal forms of expression. Thus,

the employment of effective communication strategies becomes imperative for clear idea transmission [5].

Student–teacher interaction, both within and outside the classroom, is profoundly influenced by the teaching perspective embraced by the teacher. Educators who adhere to a rhetorical perspective engage in communication with students as a means to influence or persuade them. Effective communication in this context revolves around clear teaching, the relevance of course content, and an assertive demeanor [6]. Emphasizing the significance of interaction aims to deliver meaningful lessons to all students within an inclusive educational environment [7].

The presence of teaching assistants is a phenomenon characteristic of 21st century Czech education. This development is intrinsically tied to legislative changes in 2016, specifically the amendment of Education Act No. 561/2004 and the adoption of the Decree On Inclusive Education 27/2016 Coll. These legal documents introduce the concept of support measures, with one such measure involving the utilization of teaching assistants (§ 5). Teaching assistants are tasked to provide support to other pedagogical staff in educating pupils/students with special educational needs [8]. In line with Czech legislative documents, teaching assistants have various responsibilities, including: assisting in direct educational activities by following teacher or educator instructions to support individual students, helping students achieve their educational goals and encouraging their independence, promoting the development of essential skills, hygiene, and social competencies, supporting teachers, particularly when working with students with special needs, assisting with organizational tasks related to students with special needs, aiding in the adaptation of special needs students to the school environment, facilitating communication between students, their legal representatives, and the broader community, providing necessary assistance to students in self-care and mobility during classes and school events outside the school premises, supporting the development of social competencies in students with special needs. The current scenario imposes entirely novel requirements on the teacher's role, as traditionally, they were frequently the sole presence in the classroom with their students. However, they are now thrust into a situation where collaboration with a teaching assistant becomes essential.

Consequently, communication within the classroom takes on a distinct dimension when a teaching assistant is present, as their presence introduces an additional element that can influence classroom dynamics through their conduct. DeBeck and Demaree [9] observe that teaching assistants often serve as a primary point of contact for pupils, acting as intermediaries between pupils and teachers. In this capacity, teaching assistants frequently engage in more direct interactions with students.

The role of the teacher assumes paramount importance in implementing effective learning within inclusive classrooms. In such environments, all pupils, especially those with special needs requiring tailored behavioral and engagement support, should have the opportunity to learn according to their abilities [10]. Collaboration between teachers profoundly impacts the effectiveness of inclusive classrooms and can mitigate challenges and physical hazards in the learning environment. It is imperative to gain a better understanding of the expected roles of both teachers in order to enhance inclusive education [11].

Research by Stang and Roll [12] reveals that the behaviors of teaching assistants can be predictive of student motivation and learning outcomes. Their study examines how various behaviors of teaching assistants contribute to student motivation, engagement, and overall learning outcomes.

Conversely, Bosanquet and Radford [13] underscore that teaching assistants are assuming increasing responsibilities for student learning. However, this shift can lead to a separation between these students and the teacher. Quantitative research has raised concerns about the overall impact of teaching assistant support on student progress, indicating a negative relationship between the amount of teaching assistant support provided and the progress of supported students.

In this study, we endeavor to investigate the intricate dynamics of pedagogical communication within modern classrooms, taking into special consideration the presence and impact of teaching assistants. Our primary aim is to shed light on how pedagogical communication unfolds in classrooms where teaching assistants play a pivotal role. To achieve this, we will closely examine the communication strategies and interactions between teachers, students, and teaching assistants.

We present partial results of an ongoing three-year project which seeks to (1) analyze the various dimensions of pedagogical communication, encompassing verbal and non-verbal interactions, within contemporary educational settings, and (2) to assess the distinct roles that teaching assistants assume in facilitating classroom communication and their influence on student–teacher dynamics. By pursuing these objectives, we aim to contribute to a deeper understanding of effective pedagogical communication in the modern educational landscape, with a particular focus on the role of teaching assistants. Our research endeavors to inform pedagogical practices and educational policies, ultimately enhancing the quality of education provided to all students.

The research is related to sustainability primarily by highlighting the importance of effective communication and collaboration in educational settings, which can contribute to sustainable outcomes and practices. The article discusses the inclusive practices of teaching assistants, not making a differentiation between students with special educational needs and typically developing peers. Inclusive education promotes social and environmental sustainability by fostering a sense of belonging and equity among students.

## 2. Materials and Methods

Pedagogical communication and interaction research has historically been characterized by its interdisciplinary nature, drawing upon pedagogical, psychological, linguistic, sociological, and ethnographic approaches. The synthesis of research findings from these diverse disciplines offers a more comprehensive understanding of the educational process. The primary intention of this study is to contribute to this holistic perspective by focusing on a distinct participant in the educational milieu—the teaching assistant. Education for sustainable development can be generally understood through emphasizing inclusive elements, adopting a holistic approach, and particularly recognizing the importance of communication and interaction in the educational environment.

At this juncture, we formulate the central research question: How does pedagogical communication and interaction unfold in classrooms where a teaching assistant is present? Derived from this primary inquiry, we have established the following sub-questions: (a) What are the key communication and interaction characteristics observed during teaching hours from the perspective of the teaching assistant? (b) Which categories of communication and interaction are predominant? (c) Which categories of communication and interaction are notably absent?

As our fundamental research methodology, we have opted for standardized observation. Within the realm of standardized observation, we employ the method of interaction analysis, which entails the systematic observation and assessment of communication and interaction within the classroom setting. Among the available observation systems, we have selected Flanders' interaction analysis system over A. A. Bellack's observation system [14]. While Bellack's system records individual events in a highly precise manner, including the activities of both the teacher and students, it is relatively complex, requiring the observer to monitor 54 categories meticulously designed to describe the activities occurring in the educational process [15].

Flanders' interactive approach is rooted in the concept that teaching comprises a sequence of communicative (interactive) actions that are reciprocated by both the teacher and students. Their joint participation during instruction encapsulates its defining characteristics. The term "actions" can be interchangeably replaced with "behavior categories", which can then be further refined into specific activities for clear and discernible observation and

analysis [16]. At the same time, we are aware of the limits that Flanders' method offers, as Amatari [17] states, for example.

Initially proposed by Flanders [18], a total of ten activities were identified. Over time, the number of categories expanded gradually [16].

In the development of categories for observing teaching classes featuring a teaching assistant, we have incorporated these categories as a foundation and iteratively refined them through pilot testing. Validation of these categories was conducted through in-depth analysis of four video recordings of teaching sessions by four researchers. Consequently, we have established an additional set of 16 categories focused on the activities of the teaching assistant (TA stands for teaching assistant):

TA01—The teaching assistant remains seated, monitoring the lesson's progression.

TA02—The teaching assistant sits while mentoring a student.

TA03—The teaching assistant remains seated while preparing didactic materials and instructions.

TA04—The teaching assistant stands while mentoring a student.

TA05—The teaching assistant stands, observing the teaching process without direct communication.

TA06—The teaching assistant offers assistance based on student activities.

TA07—The teaching assistant moves around the classroom without engaging in communication.

TA08—The teaching assistant moves around the classroom, providing didactic support to individual students.

TA09—The teaching assistant communicates and consults with the teacher.

TA10—The teaching assistant coordinates whole-class instruction.

TA11—The teaching assistant organizes instruction for individuals or groups.

TA12—The teaching assistant assumes teaching responsibilities in collaboration with the teacher.

TA13—The teaching assistant poses questions to the entire class.

TA14—The teaching assistant imparts new knowledge to students.

TA15—The teaching assistant evaluates and provides feedback to students.

TA16—The teaching assistant deviates from the teaching context, engrossed in their own activities.

Given the legal obligations surrounding the role of a teaching assistant, these categories do not differentiate between pupils with special educational needs (SEN) and their typically developing peers (intact pupils). Therefore, the mentoring and assistance provided by the teaching assistant are treated as inclusive practices. The teaching assistant typically occupies a designated position within the classroom, often in proximity to the assigned student, which distinguishes categories TA02 and TA04. Category TA14, where the teaching assistant imparts new knowledge to students, piques our interest in assessing the extent of their involvement in direct teaching.

To systematically analyze the video recordings and quantify the frequencies of specific interaction and communication categories, a specialized software program called Codenet Version 1.0 was utilized. This software was developed at the Department of Pedagogy and Psychology, Faculty of Education, University of Hradec Králové (authors: T. Svatoš and V. Žák). The quantitative analysis includes frequency tables of individual activity categories, a graphical overview of the teaching unit in terms of these categories, and a time-based recording of individual categories. Thus, Codenet enabled precise measurement of newly established categories within defined time intervals. Each interval was set at a duration of 3 s (i.e., it is marked as one click), allowing for granular examination of classroom communication and interactions. Four researchers actively engaged in the analysis process. Researchers selected the relevant category that best corresponded to the observed type of interaction or communication occurring in the video during each 3 s interval. To enhance the reliability and accuracy of the data, inter-rater reliability checks were conducted periodically throughout the coding process. This involved cross-checking and verifying the consistency

of coding decisions among researchers. Any discrepancies were resolved through consensus discussion, further strengthening the integrity of the dataset.

### 2.1. Selection of a Research Sample

The selection of the research sample was influenced by two fundamental factors. The first factor was the presence of a teaching assistant in the classroom and their participation in the teaching process. The second factor was the consent of the school administration and, simultaneously, the consent of the legal guardians of the students for the acquisition of video recordings. This significantly narrowed down the selection of the research sample. We collaborated with five schools, obtaining three video recordings in each school.

To ensure comprehensive analysis and robust data, 15 distinct classroom lessons were evaluated, resulting in a diverse dataset that encompassed a range of teaching scenarios. To ensure comparability and mitigate potential sources of variation, the research study focused exclusively on lessons within a specific academic domain, namely Science and Human in the 3rd grades of primary schools. Additionally, the study was conducted within a geographically concentrated area, where all participating schools were located in a single town. The selected schools exhibited relative uniformity in terms of their size and demographic composition. This deliberate choice of a homogeneous sample, characterized by uniform subject matter and geographical proximity, was made to minimize the influence of extraneous factors and geographical context, thereby enhancing the internal validity of the research findings.

Video recordings of Czech classroom lessons were employed as the primary source of data for this study. Each video recording corresponded to a single classroom lesson and had an average duration of 45 min, capturing the entirety of the instructional session.

### 2.2. The Ethical Aspects

The ethical aspects of this research were of paramount importance, and rigorous procedures were adhered to in line with established ethical standards. Informed consent was obtained from all participating educational institutions where the research was conducted. Detailed information regarding the objectives, methodologies, and potential implications of the study was provided to the schools, ensuring transparency and a clear understanding of the research's scope. By adhering to these ethical principles and securing informed consent, we aimed to conduct the research with utmost integrity, respect for the rights of all participants, and a commitment to the responsible use of collected data. This ethical framework underpins the entire research process, from data collection to analysis and dissemination of findings.

### 3. Results

Table 1 displays fundamental descriptive statistical measures for all 16 categories observed during structured video analysis. The table illustrates differences in the frequencies of individual categories, indicating the range of frequencies among these categories. The most substantial difference is observed in category TA03 where the teaching assistant sits and prepares teaching aids. TA05 and TA01 follow. These categories predominantly exhibit the passivity of the teaching assistant (e.g., for category TA01, the range was 177 with an average value of 60). These categories are accompanied by category TA07 (range 79), where the teaching assistant neither communicates nor engages in interaction, "just" moves around a classroom. At first glance, these categories indicate a rather passive approach of teaching assistants; nonetheless, teaching assistants are actively involved in monitoring pupils and observing the progress of the teaching process. In contrast, entire passivity is evident in category TA03, where the teaching assistant sits and prepares didactic instructions and materials, and in category TA16 (range 102), in which the assistant does not follow the teaching process and is engaged in their own activities which do not relate to activities of a class. To sum up, we should distinguish between forms of teaching assistant passivity. It can be perceived as non-engagement in communication and interaction within the teaching

context, but still being involved in the pedagogical process (e.g., moving around the classroom, monitoring student activities). However, another form of passivity is characterized by non-participation in teaching, focusing on personal activities without following pupils' and teacher's work (e.g., being isolated and working on non-related activities). Categories in which the assistant communicates with only one student fall within the range of 0–46. The child's identity was not distinguished; therefore, we do not know whether it was a child with special educational needs, a typically developing child, or a child assigned to a teaching assistant.

**Table 1.** Descriptive statistical measures for individual categories.

| Category | Mean | St. Deviation | Minimum | Maximum | Median | Modus |
|----------|------|---------------|---------|---------|--------|-------|
| TA01 | 60 | 47.3 | 11 | 188 | 49 | - |
| TA02 | 22 | 10.5 | 2 | 40 | 24 | 28 |
| TA03 | 50 | 55.9 | 2 | 215 | 29 | - |
| TA04 | 26 | 12.3 | 0 | 46 | 25 | 18 |
| TA05 | 55 | 44.3 | 0 | 178 | 49 | - |
| TA06 | 66 | 33.5 | 24 | 157 | 59 | - |
| TA07 | 34 | 20.7 | 13 | 92 | 32 | - |
| TA08 | 58 | 24.8 | 5 | 89 | 62 | 81 |
| TA09 | 45 | 24.4 | 0 | 87 | 45 | 62 |
| TA10 | 5 | 8.4 | 0 | 26 | 0 | 0 |
| TA11 | 20 | 12.1 | 0 | 43 | 18 | - |
| TA12 | 19 | 19.0 | 0 | 64 | 16 | 0 |
| TA13 | 2 | 2.3 | 0 | 6 | 0 | 0 |
| TA14 | 0 | 0 | 0 | 0 | 0 | 0 |
| TA15 | 17 | 13.9 | 0 | 56 | 16 | - |
| TA16 | 17 | 31.9 | 0 | 102 | 0 | 0 |

The analysis of Figure 1 provides insights into the relative prevalence of different categories of activities. Notably, the category "teaching assistant provides assistance based on student activity" exhibited the highest frequency. This implies that students proactively summon the teaching assistant, either through verbal or non-verbal cues, thus assuming an active role in initiating communication and interaction. This observation corresponds with the designated role of the teaching assistant, who is assigned to a classroom primarily to support a particular student but is also expected to contribute to the broader class dynamics and collaborate closely with the classroom teacher. Following closely is the category in which the "teaching assistant sits (either with a student requiring support measures or in their vicinity) and observes the instructional process". The third most frequently occurring category pertains to providing didactic support to individual students, denoted as "TA walks through the classroom and offers didactic assistance to individuals". This observation aligns with the statutory responsibilities of teaching assistants as outlined in relevant legislation. During the course of this study, no category was identified in which the teaching assistant introduced novel educational content. Such a role would involve the teaching assistant temporarily assuming the position of teacher and delivering new instructional material, whether to the entire class or a subgroup of students. The category involving the teaching assistant asking questions to the entire class exhibited a relatively minimal presence. However, it should be noted that this analysis did not distinguish whether the questions posed were of an organizational nature or related to the learning content. Finally, the third least prevalent category involved the teaching assistant organizing instruction for the entire class. This category distinctly underscores the extent to which teaching assistants engage in managerial support for the teacher.

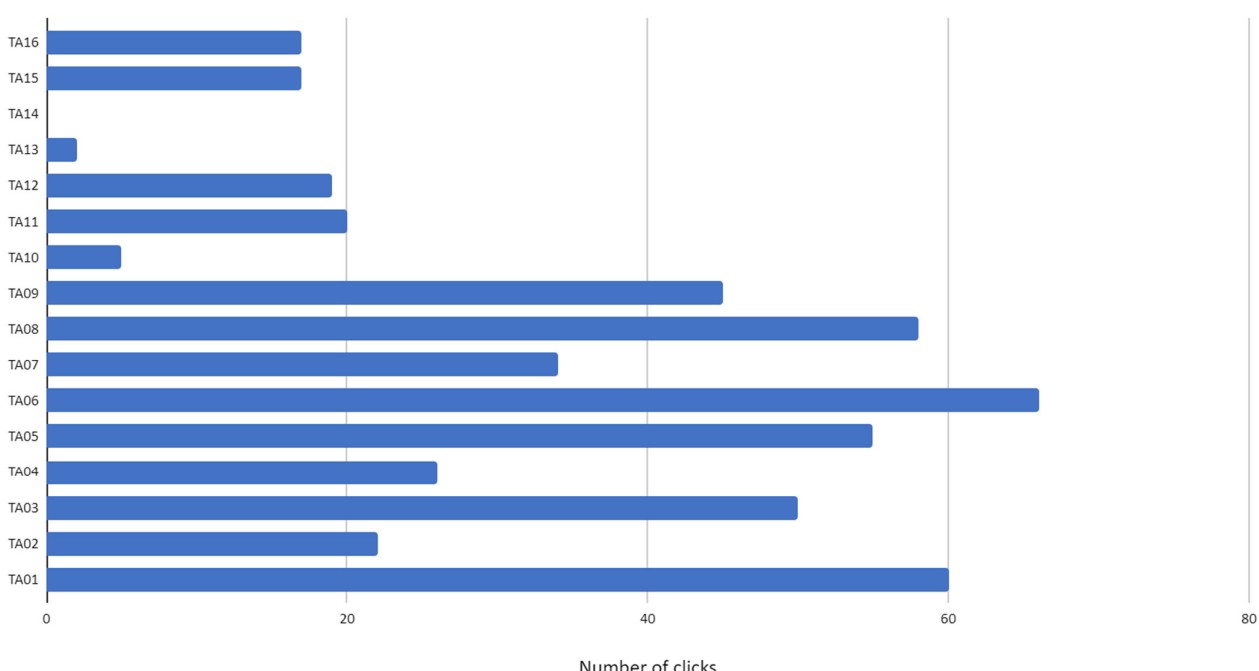

**Figure 1.** Average values of the number of intervals (interval = one click per 3 s).

To provide a more intricate portrayal of the activities undertaken by the teaching assistant during the instructional process, we present Figure 2. This graph showcases three selected instructional sessions with noteworthy variations in specific categories. In the graphical depiction of the first video analysis (lesson 1), a distinct outlier is discernible within the category in which the teaching assistant is seated and formulating didactic instructions. This category exhibits an extraordinary prevalence, while other categories are notably underrepresented. Notably, categories that require the teaching assistant to assume a prominent role in instructional leadership, such as organizing the entire class, taking charge, presenting new instructional content, assessing, and furnishing feedback, are conspicuously absent. For the second video analysis (lesson 2), we have opted to showcase it due to the elevated representation of the category where the teaching assistant assumes instructional responsibilities and collaborates with the teacher. Regrettably, the category wherein the teaching assistant orchestrates the instructional process and introduces new curricular content remains unrepresented in this instance. In the scrutiny of the third video recording (lesson 3), it is noteworthy that the category "teaching assistant provides assistance contingent upon student activity (i.e., stemming from their initiative)" is predominantly featured.

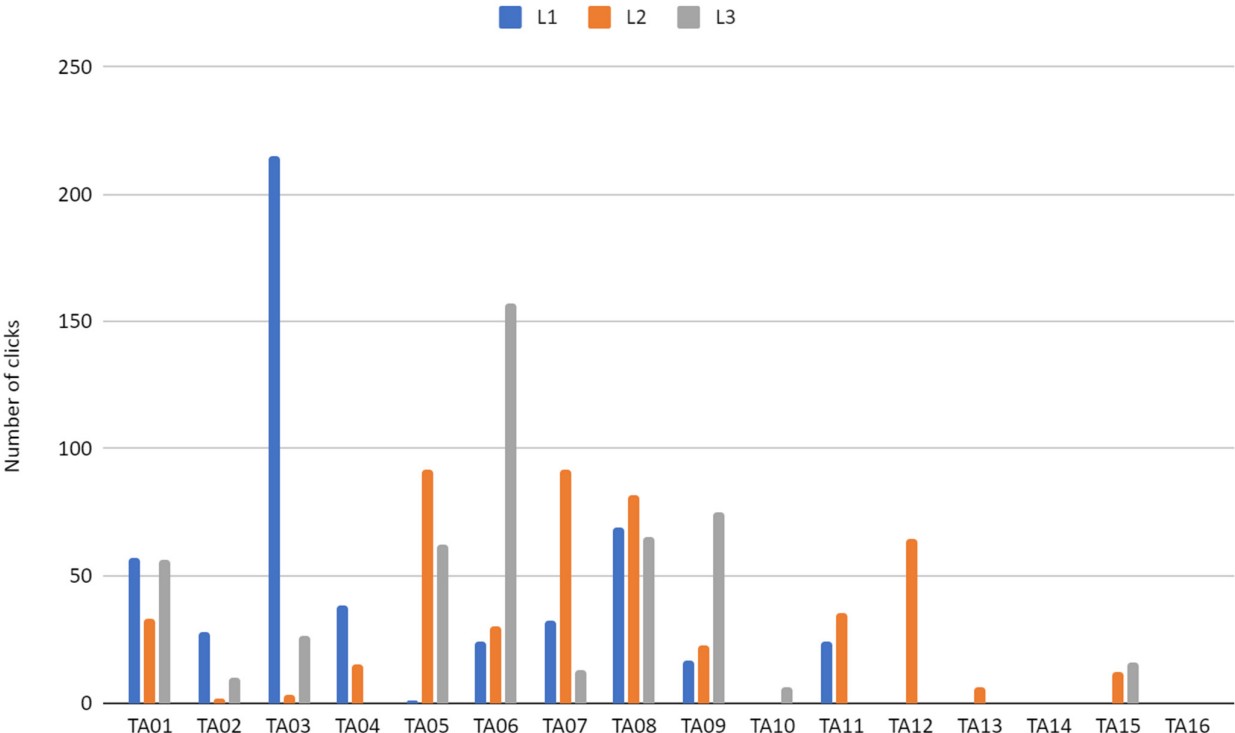

**Figure 2.** More detailed illustration of three selected lessons.

## 4. Discussion

The multifaceted role of teaching assistants within the educational context encompasses a range of responsibilities and interactions that are subject to variation and change. In this discussion, we delve deeper into the implications of the research findings by synthesizing them with insights from existing literature, as well as the provided sources. The discourse centers on key themes of guidance, training, collaboration, autonomy, and interpersonal factors, all of which have a bearing on the role of teaching assistants and the optimization of their contributions in the classroom.

Our findings resonate with the lamentable state of affairs as articulated by Russel, Webster, and Blatchford [19]. The lack of guidance for teachers in working effectively with teaching assistants has long been recognized as a systemic issue. This dearth of preparation and induction for teachers is particularly concerning, considering that nearly every teacher collaborates with teaching assistants. Therefore, a key consideration moving forward is the provision of comprehensive guidance during initial teacher education (ITE) and ongoing professional development. This need is underscored by the fact that teaching assistants are predominantly involved in activities such as preparing teaching aids, which implies a need for alignment and coordination between teachers and teaching assistants.

Furthermore, the Effective Deployment of Teaching Assistants project [19] underscores the significance of strategies that promote the active involvement of teaching assistants in lessons. While our data indicate that teaching assistants often provide support upon student initiation, their increased engagement aligns with the strategies developed and evaluated in this project. The dynamics of effective collaboration within the classroom, as identified by Logan, Bones, and Shannon [20], including teamwork, collaboration, and collegiate support, substantiate the idea that teaching assistants should play a more active role in the pedagogical process.

Notably, the role of training in the development of effective teaching assistant-teacher partnerships is highlighted by Anderson and Lyn-Cook [21]. Training, not only for teaching assistants in inclusive practices but also for teachers in the deployment of teaching assistants, emerges as a pivotal factor. Our findings reiterate the importance of well-defined roles and responsibilities to ensure equitable distribution of teacher time and prevent students from

becoming isolated. The lack of clarity in student identification, a point accentuated by our research, underscores the urgency of providing this training and developing standardized procedures for collaboration in diverse classroom settings.

Fritzsche and Köpfer [22] offer an insightful cultural comparative perspective, emphasizing the struggle for autonomy that teaching assistants face. In a field characterized by heteronomous structures, teaching assistants seek to independently structure their professional roles. Autonomy, together with formal framework conditions, profoundly shapes the role of teaching assistants. This striving for autonomy underlines the broader issue of role definition and recognition of teaching assistants within the educational ecosystem.

Intriguingly, Jardí, Webster, Petreñas, and Puigdellívol [23] elucidate the importance of interpersonal factors in building favorable partnerships between teaching assistants and teachers. Elements such as personality, gender, background, attitudes, and communication styles all play a role in fostering a positive working relationship. To ensure effective partnerships, the pairing of teachers and teaching assistants should consider these interpersonal factors, thereby promoting a harmonious and productive collaboration.

## 5. Conclusions

Based on an analysis of teaching sessions, the activities of the teaching assistant during class were comprehensively described using individual categories, identifying the most and least frequently occurring ones. The most frequently occurring categories included "Teaching Assistant provides assistance based on student/students' activity", "Teaching Assistant sits, observes the course of instruction", and "Teaching Assistant walks through the classroom, providing didactic assistance to individuals". These categories align with the legal responsibilities outlined for the teaching assistant. The least frequently occurring categories were "Teaching Assistant introduces new material" (this category was entirely absent) and "Teaching Assistant poses questions to the entire class". The analysis indicates that, in the examined video recordings, the teaching assistant did not assume the role of the teacher and did not significantly engage in more didactic activities, delegating such activities to the teacher.

Communication and interaction in teaching play a pivotal role. When an additional participant, the teaching assistant, is introduced into the instructional setting, the examination of communication and interaction becomes particularly intriguing. Our endeavor was to describe the activities of the teaching assistant based on the identified categories. The resulting database maps the frequency of each category rather than focusing on the quality of communication and interaction (it is more about "how much" than "how well"). This provides an opportunity for further research projects.

In our study, several limitations are acknowledged. The research was conducted in the Czech Republic, and therefore, the outcomes of our investigation may be influenced by the local conceptualization of the role and function of a teaching assistant within legislative frameworks. Additionally, the socio-cultural context is recognized as a potential factor that may exert an impact. Nevertheless, we posit that the Flanders Interaction Analysis System (FIAS) tool holds applicability in any European country where the role of a teaching assistant is present. Another limitation of our research pertains to the modest size of the research sample and the absence of contextual information regarding the classroom climate. It is important to note, however, that the primary aim of this study was not to make determinations about the quality of communication and interaction.

In conclusion, effective communication and interaction between teaching assistants and teachers in the context of sustainability education can help foster a holistic and integrated approach to teaching, learning, and practicing sustainability. It empowers students with the knowledge and skills needed to contribute to a more sustainable future and creates a culture of responsibility and awareness within educational institutions and the broader community.

**Author Contributions:** Conceptualization, M.M., J.W., M.S. and J.V.; methodology, M.M. and J.V.; formal analysis, M.M. and J.W.; resources, J.W. and M.S.; data curation, M.M. and J.V.; writing—M.M., J.W., M.S. and J.V.; writing—M.S. and J.V. All authors have read and agreed to the published version of the manuscript.

**Funding:** This research was funded by the University of Hradec Králové. The presented results are a part of a specific three-year project No 2105/2022 Pedagogical communication and interaction in a classroom with a teacher assistant.

**Institutional Review Board Statement:** The study has been approved by the Committee for Research Ethics at the University of Hradec Králové, no. 7/2023 being in accordance with the Ethical Research Framework of the Ministry of Education, Youth and Sport, Czechia, and ethical requirements in research.

**Informed Consent Statement:** All the subjects involved in the study were informed about the new pieces of content they learned which were permitted by the school principal.

**Data Availability Statement:** All data can be obtained from the corresponding author.

**Conflicts of Interest:** The authors declare no conflicts of interest.

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
