# Peer review of "Communication and Interaction Practices in Czech Classrooms with a Teaching Assistant"

_sustainability, doi:10.3390/su16030989_

Round 1

Reviewer 1 Report

Comments and Suggestions for Authors

While the work is interesting, it lacks a theoretical framework to support the research questions posed.

The methodology is still weak, and many aspects remain unclear and need to be clarified.

The results and conclusions are weak.

Author Response

Dear Sir or Madam,

We appreciate your thoughtful review, and as per your feedback, the text has been duly revised.

Yours faithfully,

The Author Team.

Reviewer 2 Report

Comments and Suggestions for Authors

Paper deals with important educational topic of the roles and influence of teaching assistants on classroom communication, emphasizing their impact on student-teacher interactions. Study is conducted in Czech educational system, but research results and conclusions can be can be applied to others European and similar educational systems. Paper contributes to better understanding of pedagogical communication and interaction and results can be used for improvement of educational practices.  Therefore, this work can be significant for educational practitioners and researchers in different areas and settings.

Before publishing it is recommended that author make some changes in the manuscript:

-          Abstract should be shortened and focused on description on problem, aim, goals, methods, results, implications and limitations. No need for general statements and observations.

-          Section Materials and Methods should be shortened and focused on key methodological explanation, for example standardized observation, with minimal theoretical explanations.

-          Limitations of the research should be mentioned in the abstract and conclusion section

-          It is recommended that authors separate Conclusion section at the end and in that section describe in short manner key results, implications, possibilities for new researches, concerns and limitations.

Comments on the Quality of English Language

/

Author Response

Dear Sir or Madam,

We extend our gratitude for your profoundly insightful comments. The text has been revised in accordance with your feedback.

Yours faithfully,

The Author Team.

Reviewer 3 Report

Comments and Suggestions for Authors

The aim of the authors is to investigate the dynamics of pedagogical communication within modern Czech classrooms, taking into special consideration the presence and impact of teaching assistants.

The paper topic is interesting but needs to be revised.

- The sub-questions established by the authors in lines 135-141 should be:

(a) What are the key communication and interaction characteristics observed during teaching hours from the perspective of the teaching assistant?

(b) Which categories of communication and interaction are predominant?

(c) Which categories of communication and interaction are notably absent?

(b) and (c) are related, however they are different sub-questions

 - In line 196 the authors refer to four videos - "through in-depth analysis of four video recordings of teaching sessions", however in the study carried out only three videos were considered

 - In line 227 the authors mention “… which distinguishes categories A02 and A04. Category A14,…” the categories A02, A03 and A14 are not defined in the text. Did the authors mean TA02, TA04 and TA14?

- On line 321 there is a typo "AP walks through the classroom and offers didactic assistance to individuals." - instead of AP it should be TA.

 - Throughout the text, the authors refer to "teacher assistant" and "pedagogical assistant". Is it the same subject or does it refer to different subjects? This must be explained

 - The Achilles' heel  of this work is: the table and the graphs.

- The study that the authors show in table 1 should be better explained in the text. The authors only mention the frequency. Graph 1 is a complement to the study carried out in table 1.

- Graph 2 is not clear. What is L1, L2 and L3? perhaps the authors wanted to refer to lesson 1, lesson2 and lesson3 but this is not written in the text nor made explicit in the graph caption

- What is the study shown in graph 2? what do the vertical axis mean (what are the numbers 0, 50, ... , 250 related to?)

Comments on the Quality of English Language

-

Author Response

(The authors gave the same response as above.)

Round 2

Reviewer 3 Report

Comments and Suggestions for Authors

The authors took into account the comments and suggestions given. They revised and changed the text accordingly.

Author Response

Dear Sir or Madam,

we express our gratitude for your insightful remarks, which proved instrumental in refining the manuscript. In response to your suggestions, we have incorporated an elucidating commentary within the methodology section.

Yours faithfully

autors
